# Biofilms Positively Contribute to *Bacillus amyloliquefaciens* 54-induced Drought Tolerance in Tomato Plants

**DOI:** 10.3390/ijms20246271

**Published:** 2019-12-12

**Authors:** Da-Cheng Wang, Chun-Hao Jiang, Li-Na Zhang, Lin Chen, Xiao-Yun Zhang, Jian-Hua Guo

**Affiliations:** 1Department of Plant Pathology, Key Laboratory of Monitoring and Management of Crop Diseases and Pest Insects (Ministry of Agriculture), Engineering Center of Bioresource Pesticide in Jiangsu Province, College of Plant Protection, Nanjing Agricultural University, Nanjing 210095, China; dacheng0819@126.com (D.-C.W.); chjiang@naju.edu.cn (C.-H.J.); 2016202015@njau.edu.cn (L.-N.Z.); 2017102018@njau.edu.cn (L.C.); 2Key Laboratory of IPM on Crops in Northern Region of North China (Ministry of Agriculture), Institute of Plant Protection, Hebei Academy of Agricultural and Forestry Science, Integrated Pest Management Center of Hebei Province, Baoding 071000, China; zxy_zxl@163.com

**Keywords:** *Bacillus amyloliquefaciens*, biofilm formation, drought tolerance, stomatal aperture, abscisic acid

## Abstract

Drought stress is a major obstacle to agriculture. Although many studies have reported on plant drought tolerance achieved via genetic modification, application of plant growth-promoting rhizobacteria (PGPR) to achieve tolerance has rarely been studied. In this study, the ability of three isolates, including *Bacillus amyloliquefaciens* 54, from 30 potential PGPR to induce drought tolerance in tomato plants was examined via greenhouse screening. The results indicated that *B. amyloliquefaciens* 54 significantly enhanced drought tolerance by increasing survival rate, relative water content and root vigor. Coordinated changes were also observed in cellular defense responses, including decreased concentration of malondialdehyde and elevated concentration of antioxidant enzyme activities. Moreover, expression levels of stress-responsive genes, such as *lea*, *tdi65*, and *ltpg2*, increased in *B. amyloliquefaciens* 54-treated plants. In addition, *B. amyloliquefaciens* 54 induced stomatal closure through an abscisic acid-regulated pathway. Furthermore, we constructed biofilm formation mutants and determined the role of biofilm formation in *B. amyloliquefaciens* 54-induced drought tolerance. The results showed that biofilm-forming ability was positively correlated with plant root colonization. Moreover, plants inoculated with hyper-robust biofilm (Δ*abrB* and Δ*ywcC*) mutants were better able to resist drought stress, while defective biofilm (Δ*epsA-O* and Δ*tasA*) mutants were more vulnerable to drought stress. Taken altogether, these results suggest that biofilm formation is crucial to *B. amyloliquefaciens* 54 root colonization and drought tolerance in tomato plants.

## 1. Introduction

Tomato plants (*Solanum lycopersicum*), as an important part of the agricultural industry, are widely cultivated across the world. World production of tomatoes was 177 million tons in 2016. Abiotic stresses are various adverse environmental factors that cause severe declines in yields of tomato and other plant crops. In terms of plant physiology, drought is a common stress environmental factor that limits growth and leads to mortality of crops. Moreover, due to global warming, drought stress has become an urgent issue in agricultural research [1]. Therefore, clarifying potential regulatory mechanisms is key in mitigating losses from drought stress.

As we know, plants sense and respond to abiotic stress through various signaling molecules. For example, the plant hormone abscisic acid (ABA) is synthesized under dehydration conditions and plays a vital role in transducing stress signals [2]. Under drought stress conditions, ABA rapidly elicits plant reactions such as stomatal closure, alleviating water loss by limiting transpiration [3]. To avoid potential damage, plants have developed various antioxidant enzymes, including superoxide dismutase (SOD), peroxidase (POD), catalase (CAT) and ascorbate peroxidase (APX), which function as ROS scavengers [4,5,6,7].

Plant growth-promoting rhizobacteria (PGPR) refers to a beneficial and heterogeneous group of microbes that enhance plant growth and productivity through a wide variety of mechanisms [8]. Liu et al. reported that PGPR has a stable ability of rhizosphere colonization and promotes plant growth by enhancing plant nutrient absorption, improving soil structure, or increasing water-holding capacity [9]. Besides, PGPR strains promote induced systemic tolerance (IST) in plants to different abiotic stresses, such as drought [10,11], salt [12,13], heat [14,15], and cold stress [16,17]. Furthermore, PGPR strains reportedly enhance disease resistance to many plant pathogens, such as *Pseudomonas syringae* pv. tomato DC3000 [18,19,20], *Ralstonia solanacearum* [21], and *Botrytis cinerea* [22,23]. *Bacillus* species are among the most frequently isolated endophytes and some of them have the ability to increase crop yield and suppress plant disease [24]. The potential mechanisms include increased nutrient uptake, phytohormone production synthesis, and volatiles synthesis [25,26]. Furthermore, they have been reported to alleviate the negative effects of various abiotic stresses [27].

Biofilms are defined as self-organized, cooperating communities in which microbial cells adhere to each other on living or non-living surfaces within an extracellular matrix produced by themselves and can be prevalent in natural, clinical, and industrial settings. Most microorganisms associate with biotic and abiotic surfaces through biofilms [28]. Moreover, previous studies have demonstrated that PGPR biofilm formation plays a crucial role in protecting plants from abiotic and biotic stress. PGPR are frequently present in the rhizosphere and form biofilm-like structures on plant roots to protect against stress. For instance, *Pseudomonas putida* strains colonize root surfaces and produce biofilm-like structures that protect against drought stress [11]. *Bacillus amyloliquefaciens* biofilm formation improves salt stress tolerance in barley [29]. Furthermore, *Bacillus subtilis* forms biofilm-like structures that protect against tomato wilt disease [30]. Although some studies have been published, there is little direct evidence illustrating the role of biofilm formation in protection against abiotic and biotic stresses, especially drought stress.

The genetic regulatory circuit that governs biofilm formation has been well studied [31,32,33]. Two key matrix gene operons, *tapA-sipW-tasA* and *epsA-O*, are involved in synthesis of two key matrix components, protein fibers (TasA) and exopolysaccharides (EPS), respectively, and are directly regulated by the biofilm master repressor SinR [34,35]. Spo0A~P functions as a master regulatory protein that regulates expression of the sinI gene, which encodes an anti-repressor of SinR [34,36,37,38]. Multiple histidine kinases sense various environmental signals, regulating Spo0A through protein phosphorylation [39,40,41]. AbrB, another repressor that represses the two matrix gene operons, is also negatively regulated by Spo0A~P (Figure 1D) [42,43]. Additionally, a Spo0A~P-independent regulatory pathway that includes YwcC and SlrA controls matrix gene expression. SlrA is inhibited by YwcC and contributes to biofilm formation by repressing SinR [44].

In this study, we compared the impact of root inoculation with wild-type *B. amyloliquefaciens* 54 and its hyper-robust biofilm (Δ*abrB* and Δ*ywcC*) and defective biofilm (Δ*epsA-O*, and Δ*tasA*) mutants on drought tolerance in tomato plants.

## 2. Results

### 2.1. Beneficial Rhizobacteria Significantly Improve Drought Tolerance

Of the bacterial isolates, three showed strong drought tolerance, with survival rates exceeding 40% after the water deprivation period, while control bacteria without PGPR root inoculation had a survival rate of only 6.25% (Appendix A). Of these three isolates, *B. amyloliquefaciens* 54 conferred the strongest drought tolerance, and the survival rate of treated seedlings reached 61.54%.

### 2.2. B. Amyloliquefaciens 54 Forms Robust Biofilms Both in LBGM Medium and on The Surface of Tomato Roots

Previous studies have mentioned that the bacteria can form multicellular biofilm communities that are crucial to efficient plant root colonization and can protect plant roots against environmental hazards [45,46]. The strong drought tolerance induced by *B. amyloliquefaciens* 54 could be related to formation of biofilms. Therefore, we evaluated its biofilm formation by monitoring pellicle characteristics and colony morphologies in bofilm medium LBGM (lysogenic broth [LB] supplemented with 1% glycerol and 100 μM MnSO4). As expected, *B. amyloliquefaciens* 54 formed robust and wrinkled pellicles in LBGM liquid medium (Figure 1A,C) and complex colony patterns on agar plates (Figure 1B). According to the regulatory genetic circuitry that regulates biofilm formation (Figure 1D) [31,32,33], null mutations of these genes which result in hyper-robust biofilms (Δ*abrB* and Δ*ywcC*) and strong defective biofilms (Δ*epsA-O* and Δ*tasA*) were introduced into wild-type *B. amyloliquefaciens* 54. The pellicle formations and colony morphologies of the wild-type strain and its derivatives were examined in defined media (LBGM). As shown in Figure 1, compared to wild 54, null mutations in *abrB* or *ywcC* caused formation of hyper-robust biofilm pellicles (Figure 1A,C) in LBGM liquid medium and more complex colony patterns (Figure 1B) on LBGM agar plates, whereas *epsA-O* or *tasA* mutants formed pellicles with unwrinkled or featureless surfaces and colony patterns with apparent defects (Figure 1A–C). These results suggest that these genes are related to *B. amyloliquefaciens* 54 biofilm formation.

Considering that *B. amyloliquefaciens* 54 forms strong biofilms in defined medium, we wondered whether it might also form biofilms on the surface of plant roots. Therefore, we examined biofilm formation of *B. amyloliquefaciens* 54 and its biofilm mutants on tomato plant roots. We introduced a constitutively expressed GFP (Green Fluorescent Protein) reporter into the wild-type strain and its biofilm mutants to better view the biofilms on roots. As expected, wild-type *B. amyloliquefaciens* 54 colonized the roots and formed strong biofilms, and Δ*abrB* and Δ*ywcC* mutants formed larger cell clusters on the roots and more obvious biofilm-like structures (Figure 2). Conversely, knockout of *epsA-O* or *tasA* induced a sharp decrease in colonization, and almost no biofilm formation was detected on the roots (Figure 2).

### 2.3. Biofilm Formation Contributed to B. Amyloliquefaciens 54-Induced Drought Tolerance

To reveal whether biofilm formation regulates drought stress tolerance, we root-inoculated wild-type *B. amyloliquefaciens* 54 and its biofilm mutants. After being deprived of water for 15 days before being watered again, plants treated with hyper-robust biofilm mutants (Δ*abrB* and Δ*ywcC*) showed stronger drought tolerance than that displayed by plants treated with wild-type *B. amyloliquefaciens* 54, while plants treated with defective biofilm mutants (Δ*epsA-O* and Δ*tasA*) showed weaker drought tolerance (Appendix A).

The relative water content (RWC) of leaves is a crucial indicator of the water status in plants and plays a vital role in protecting plants from over-dehydration stress. Although root inoculation with wild-type *B. amyloliquefaciens* 54 and all tested biofilm mutants improved RWC over that of control plants, hyper-robust biofilm mutant (Δ*abrB* and Δ*ywcC*) treatment increased the RWC of leaves more than did wild-type treatment, whereas defective biofilm mutant (Δ*epsA-O* and Δ*tasA*) treatment resulted in lower leaf RWC (Figure 3A).

A previous study showed that root vigor is a sharp and reliable biomarker of plant drought stress tolerance. Similar to the RWC results, all *B. amyloliquefaciens* 54 treatments improved root vigor over that of control plants. Hyper-robust biofilm mutant (Δ*abrB* and Δ*ywcC*) treatment, especially with the Δ*abrB* mutant, resulted in higher root vigor that did wild-type treatment, while defective biofilm mutant (Δ*epsA-O* and Δ*tasA*) treatment resulted in lower root vigor (Figure 3B).

Abiotic stress is known to promote peroxidation of membrane lipid and electrolyte leakage in plant tissues, and malondialdehyde (MDA) is frequently used as an indicator of lipid peroxidation. Therefore, leaf MDA content was determined. The data indicated that wild-type *B. amyloliquefaciens* 54 and all mutants reduced leaf MDA content. Furthermore, hyper-robust biofilm mutant (Δ*abrB* and Δ*ywcC*) treatment reduced MDA content more than did wild-type treatment, while defective biofilm mutant (Δ*epsA-O* and Δ*tasA*) treatment reduced MDA content (Figure 3C). The above results suggest that *B. amyloliquefaciens* 54 biofilm formation is positively related to drought stress tolerance in tomato plants.

### 2.4. Biofilm Formation Was Conductive to B. Amyloliquefaciens 54-Regulated Stress-Related Genes

To further elucidate the role of *B. amyloliquefaciens* 54 in drought stress, we analyzed the expression of several stress-related genes, including *lea*, *ltpg2*, and *tdi-65*. As shown in Figure 3, pre-treatment with wild-type *B. amyloliquefaciens* 54 increased expression of these genes relative to expression after mock treatment (Figure 3D–F). To clarify whether biofilm formation is involved in the regulatory mechanism, we inoculated plants with biofilm mutants of *B. amyloliquefaciens* 54. As shown in Figure 3, compared with wild-type *B. amyloliquefaciens* 54 inoculation, hyper-robust biofilm mutant (Δ*abrB* and Δ*ywcC*) treated plants produced higher gene expression, while defective biofilm mutant (Δ*epsA-O* and Δ*tasA*) treatment produced lower gene expression. This suggests that *B. amyloliquefaciens* 54 biofilm formation is positively involved in the observed induction of stress-related gene expression.

### 2.5. Biofilm Formation Was Involved in B. Amyloliquefaciens 54-Mediated Antioxidant Enzyme Activities

Reactive oxygen intermediates (ROIs) are widely used signaling molecules that control various processes, such as pathogen defense, abiotic stress responses and programmed cell death [47]. SOD, CAT, POD, and APX are referred to as the major ROI-scavenging enzymes. Therefore, we investigated the activities of these antioxidant enzymes in the leaves of tomato plants pre-inoculated with *B. amyloliquefaciens* 54 and its biofilm mutants. Under drought stress, *B. amyloliquefaciens* 54 treatment elevated the activities of SOD, CAT, POD, and APX of tomato plants at 7 and 11 days compare to mock-inoculated controls (Figure 4A–C and Figure 3D). Furthermore, hyper-robust biofilm mutant (Δ*abrB* and Δ*ywcC*) treatment elevated SOD, POD, CAT, and APX activities compared to wild-type treatment at 11 days and 7 days (Figure 4A–C and Figure 3D). In addition, defective biofilm mutant (Δ*epsA-O* and Δ*tasA*) treatment decreased SOD and POD activities compare to wild-type treatment at 11 days (Figure 4A,B). However, CAT and APX activities displayed similar trends between the defective biofilm mutant (Δ*epsA-O* and Δ*tasA*) treatment and wild-type treatment (Figure 4C and Figure 3D).

### 2.6. Biofilm Formation Was Involved in B. Amyloliquefaciens 54-Induced Stomatal Closure in Tomato Plants

Root-associated *B. subtilis* restricts stomatal openings in response to drought and pathogen challenges [48]. To clarify whether *B. amyloliquefaciens* 54 induces stomatal closure under drought stress conditions, we pre-treated tomato plant roots with *B. amyloliquefaciens* 54 and measured the stomatal apertures of the epidermis via microscopy. As shown in Figure 5, root inoculation with *B. amyloliquefaciens* 54 reduced stomatal apertures at 8 and 24 h compared to that in mock-inoculated controls (Figure 5A,B). Furthermore, both hyper-robust biofilm mutant (Δ*abrB* and Δ*ywcC*) and defective biofilm mutant (Δ*epsA-O*, and Δ*tasA*) treatments induced stomatal closure (Figure 5A,B). Interestingly, both 8 h and 24 h post-inoculation, plants treated with hyper-robust biofilm mutants (Δ*abrB* and Δ*ywcC*), especially with the Δ*abrB* mutant, showed smaller stomatal sizes than did those inoculated with the wild-type; on the other hand, those treated with defective biofilm mutants (Δ*epsA-O* and Δ*tasA*), especially with the Δ*tasA* mutant, showed larger apertures (Figure 5A,B). These results indicate that *B. amyloliquefaciens* 54 induced stomatal closure, with robust biofilm mutants showing enhanced ability and defective biofilm mutants showing decreased ability to mediate such closure.

### 2.7. ABA Is Required for B. Amyloliquefaciens 54-Mediated Stomatal Closure

Under drought stress, plants accumulate the hormone ABA, which plays a major role in stomatal closure to prevent water loss by transpiration [49,50,51]. Therefore, to investigate whether ABA contributes to *B. amyloliquefaciens* 54-triggered stomatal closure, we examined the transcript levels of the *nced1* gene that encodes 9-cis-epoxycarotenoid dioxygenase, which is a key enzyme in ABA biosynthesis, and the ABA contents in tomato leaves 8 and 24 h post-inoculation with wild-type *B. amyloliquefaciens* 54 and its biofilm mutants. The results showed a significant increase in *nced1* gene transcription (Figure 5C) and ABA content (Figure 5D) compared to those in non-inoculated controls. Furthermore, hyper-robust biofilm mutant (Δ*abrB* and Δ*ywcC*) treated plants displayed higher *nced1* gene expression and ABA content than did wild-type-treated plants, whereas defective biofilm mutant (Δ*epsA-O* and Δ*tasA*) treated plants showed lower nced1 gene expression and ABA content. These findings demonstrated that ABA is involved in *B. amyloliquefaciens* 54-mediated stomatal closure and that biofilm formation contributes to ABA biosynthesis regulation by *B. amyloliquefaciens* 54.

## 3. Discussion

In recent years, the functions of PGPR in improving plant health by promoting resistance to pathogens, insect pests, and abiotic stressors, such as drought and salinity, have been well illustrated [10,11,13]. *B. amyloliquefaciens* and other members of the genus *Bacillus* have been implicated in a range of plant functions from plant growth and development to biotic and abiotic stress tolerance. In this study, we isolated wild strains of *B. amyloliquefaciens* 54, which has been demonstrated to enhance drought tolerance. However, how this strain increases drought tolerance in the rhizosphere is still not well understood.

Biofilms encase populations of different organisms and are pivotal to colonization and maintenance of long-term associations between microbes and plant hosts [52,53]. Biofilms protect against environmental stressors [54,55]. Timmusk et al. showed that biofilm formation ability of *Paenibacillus polymyxa* contributes to enhancing wheat drought tolerance [56]. Lu et al. showed that the EPS-encoding gene *epsc* plays a vital role in *Bacillus amyloliquefaciens* FZB42 to boost the drought tolerance of *Arabidopsis thaliana* [57]. In our study, we paid attention to the function of biofilm formation in plant drought tolerance and present several pieces of evidence suggesting its importance. First, we demonstrated that the isolated wild-type *B. amyloliquefaciens* 54, which has an excellent ability to help tomato plants against drought stress, forms robust biofilms (Figure 1A–C). Furthermore, we applied its biofilm mutants to demonstrate that its biofilm formation ability is positively related to colonization ability and drought tolerance efficacy (Figure 2). That is, hyper-robust biofilm mutants significantly improved colonization ability and drought tolerance efficacy, while defective biofilm mutants showed substantially decreased colonization ability and drought tolerance efficacy compared to the same parameters in plants treated with the wild-type strain. These results demonstrate that the capacity of *B. amyloliquefaciens* 54 to form biofilms in LBGM medium is positively related to its capacity to do so on tomato plant roots.

Leaf RWC was introduced as a criterion of a plant’s water status that accurately presents the balance between water absorption through root hairs and water consumed through transpiration [58]. The present study showed that tomato plants pre-inoculated with *B. amyloliquefaciens* 54 had relatively higher RWC under drought stress conditions than did untreated controls (Figure 3A). These results confirm previous reports, in which PGPR was shown to increase RWC of plants [59,60]. Furthermore, the biofilm formation ability was positively related to leaf RWC; hyper-robust biofilms increased RWC, while defective biofilms decreased RWC compared to that in plants treated with the wild-type strain. The results demonstrate that biofilm formation is involved in *B. amyloliquefaciens* 54-induced RWC of tomato plants.

Root vigor has been widely used as a sharp and reliable biomarker of drought tolerance and reflects the capacity of water and nutrition absorption under drought stress conditions [61]. A previous study showed that a consortium of three PGPR strains could increase root vigor and promote drought tolerance in cucumber plants [4]. In our study, *B. amyloliquefaciens* 54-treated plants displayed sharply increased root vigor compared with mock-inoculated controls during the drought stress period. In addition, hyper-robust biofilms increased root vigor, while defective biofilms decreased root vigor compared to that of plants treated with the wild-type strain. This suggests that biofilm formation contributes to *B. amyloliquefaciens* 54-induced root vigor enhancements.

A general component of the plant response to drought stress is the accumulation of ROS compounds which can harm the production of biomolecules, such as lipids, nucleic acids, and proteins, and which therefore need to be neutralized [62,63]. Peroxidation of membrane lipids is a major damaging effect of ROS. MDA is a widely used indicator of oxidative lipid injury in plants [64]. In our investigation, leaf MDA content was significantly lower in plants pre-inoculated with *B. amyloliquefaciens* 54 (Figure 3C). The result is consistent with previous studies, in which PGPR was shown to decrease the MDA content of plants under abiotic stress conditions [65,66]. Moreover, hyper-robust biofilms increased leaf MDA content, while defective biofilms decreased leaf MDA content compared to that in plants treated with the wild-type strain (Figure 3C), suggesting that *B. amyloliquefaciens* 54 biofilm formation protects cell membranes from ROS and decreases lipid peroxidation, thereby enhancing drought tolerance. In addition, to prevent oxidative stress and cellular damage, plants produce ROS-scavenging enzymes, including SOD, POD, CAT, and APX [12,67]. Under drought stress conditions, plants pre-inoculated with *B. amyloliquefaciens* 54 displayed relatively higher SOD, POD, CAT, and APX activities than did mock-inoculated controls (Figure 4A–D). Furthermore, hyper-robust biofilms increased these activities, while defective biofilms decreased the activities relative to those in plants treated with the wild-type strain (Figure 4A–D). These results indicate that *B. amyloliquefaciens* 54 induced antioxidant enzyme activities, with robust biofilm mutants showing enhanced ability and defective biofilm mutants showing decreased ability to mediate such activities.

Stomatal closure is the primary response to drought stress in most plants that curtail water loss from transpiration. Therefore, the stomatal aperture is a major factor that contributes to drought tolerance [68,69]. The present study demonstrated that plants pre-inoculated with *B. amyloliquefaciens* 54 had smaller stomatal aperture than did control plants (Figure 5A,B). The results confirm previous reports that PGPR could induce stomatal closure [70]. In addition, biofilm formation was positively related to stomatal aperture. That is, hyper-robust biofilms decreased stomatal aperture, whereas defective biofilms increased stomatal aperture relative to that in plants treated with the wild-type strain.

ABA was reported as a central regulator of stomatal development and aperture [71,72]. Previous studies reported that water shortage in plant roots would increase more pronounced ABA biosynthesis levels in roots than in leaves. However, ABA produced in leaf tissues plays a more important role than ABA produced in root tissues, although root ABA was also transported to leaf tissues to maintain stomatal closure [64]. Thus, *nced1*, an ABA biosynthesis gene, and leaf ABA content were measured. *B. amyloliquefaciens* 54 treatment significantly increased *nced1* gene expression and ABA content compared with levels in uninoculated controls. These results were in agreement with previous studies suggesting that PGPR could induce ABA biosynthesis [70,73]. Additionally, hyper-robust biofilms increased both *nced1* gene expression and ABA content, while defective biofilms decreased both compared to levels in plants treated with the wild-type strain.

In summary, our results demonstrated that *B. amyloliquefaciens* 54 is able to form biofilms on the surface of plant roots and that biofilm is positively associated with *B. amyloliquefaciens* 54-induced drought stress tolerance. It also provides a new theoretical basis for plant drought resistance induced by PGPR and new theoretical support for the application of PGPR in arid environments. Besides, we hypothesize that chemical communication is a crucial link between the plant roots and bacterium. Chen et al. showed that *B. subtilis* can sense the chemical signal released from plant roots and induce biofilm formation [74]. Therefore, it will be interesting to identify those chemical signal molecules that trigger *Bacillus* sp. to form biofilm and to understand the mechanism how the bacterium senses and responds to the signal in regulating biofilm formation.

## 4. Materials and Methods

### 4.1. Bacterial Strains and Culture Conditions

All 30 examined bacterial strains in this study came from our biocontrol bacteria library, which consisted of beneficial bacteria isolated by our lab from the field. They were cultured at 28 °C for 24 h on Luria–Bertani (LB) agar medium. A single colony from a freshly streaked plate was then selected and inoculated into LB broth and incubated at 28 °C for 16 h in a shaker at 200 rpm. The broth culture was spun at 6000× *g* in a centrifuge for 15 min, and the resulting pellet was resuspended in sterile water and adjusted to a final concentration of 10^6^ colony forming units (CFU)/mL (OD_600_ = 0.5) for further experiments.

Genomic DNA from *B. subtilis* 3610 mutants was used to construct corresponding *B. amyloliquefaciens* 54 mutants according to the established protocol for transformation of *Bacillus* spp. [75]. To view the biofilms surrounding the roots of tomato plants, the genomic DNA of *B. subtilis* strain YC21 was used to construct a GFP fusion containing a constitutive expression GFP reporter according to a method described previously [30].

### 4.2. Plant Growth and Treatments

Tomato seeds (*S. lycopersicum* cv. Shanghai 903) were sterilized for 30 s with 95% (*v*/*v*) ethanol followed by a further 5 min in 5% (*w*/*v*) NaClO, and then washed with sterile distilled water three times. Afterwards, the seeds were spread onto solid Murashige and Skoog (MS) medium to germinate. The seedlings were transplanted into natural clean soil and incubated in a greenhouse with a relative humidity of 60% and a 16/8 h day/night photoperiod. The tomato plantlets were transplanted into pots after two weeks. One week later, 20 mL of each adjusted suspension was applied to the plants by root irrigation. Sterile distilled water was applied as a control in this experiment. For each treatment, 24 seedlings were applied. The plants were then deprived of water for 15 days as drought stress treatment. After 15 days, normal watering was resumed. To characterize drought tolerance, survival rate of each treatment was measured after 5 days rewatering.

### 4.3. Biofilm Formation Assay

To evaluate biofilm formation of wild-type *B. amyloliquefaciens* 54 and its derivatives, we monitored colony morphology on LBGM agar plate and pellicle formation on LBGM liquid media. Cells were first grown in 3 mL LB broth to late exponential growth phase. For colony morphology, 2 μL of LB pre-culture (OD_600_ = 0.5) was spotted onto LBGM agar plates, which were incubated at 30 °C for 36 h prior to analysis. To analyze pellicle biofilm formation, 10 μL of LB pre-culture (OD_600_ = 0.5) was added to 10 mL of LBGM liquid medium and cultured for 48 h at 30 °C. Images were taken by a Canon digital camera (EOS Digital 600D, Canon Co, Tokoyo, Japan).

### 4.4. Biofilm Formation of B. Amyloliquefaciens 54 and Its Biofilm Mutants on Tomato Roots

Wild-type *B. amyloliquefaciens* 54 and its derivatives that contained *gfp* reporters were used to view biofilm formation on the root surfaces of tomato plants. Sterilized tomato seeds were grown on solid MS medium for 4 days at 25 °C. Tomato seedlings were then transplanted into MS liquid medium for 48 h at 25 °C with shaking at 60 rpm. Microbe suspensions were then added into the medium with initial OD_600_ = 0.2. After a further 48 h culture, the tomato roots were examined for biofilm formation with confocal laser scanning microscopy (CLSM; LSM710 microscope; Carl Zeiss AG, Oberkochen, Germany).

### 4.5. Stomata Assays

To measure the stomatal apertures of tomato plants pre-treated with *B. amyloliquefaciens* 54 or its derivatives, fully expanded tomato leaves were detached, and then incubated in MES buffer (pH 6.2) for detection [76]. Stomatal aperture images were recorded under microscopy (Zeiss Axio Observe 3m and Zen Standard Software, Carl Zeiss AG, Oberkochen, Germany). For each treatment, at least 60 randomly selected stomata were measured in triplicate.

### 4.6. Leaf ABA Content Determination

To determine ABA content, 1 g fresh leaf tissue was soaked in liquid nitrogen, and then extracted in 80% (*v*/*v*) methanol. The extract was centrifuged at 4000× *g* for 20 min twice, and the supernatant was recovered and evaporated under a vacuum to remove the methanol. Afterwards, ethyl acetate was added to combine with the organic fraction at pH 3.0. Under a vacuum, the dry combined fractions were extracted and dissolved in Tris-buffered saline (150 mM NaCl, 1 mM MgCl2, 50 mM Tris–HCl, pH 7.8), and ABA content was measured by an enzyme-linked immunosorbent assay (ELISA) kit (Agdia, Elkhart, IN, United States) [70].

### 4.7. RNA Isolation, RT-PCR, and qRT-PCR

The total RNA of tomato plant leaves was isolated using a Plant RNA Kit (Tiangen Bio-Tech, Beijing, China). The first cDNA strand from 1 μg total RNA was synthesized using a PrimeScript RT reagent kit (TaKaRa Bio Inc., Kyoto, Japan) and oligo (dT) primers. qRT-PCR was conducted on an ABI 7500 system (Applied Biosystems, Foster City, CA, USA) with the SYBR premix ExTaq mixture (TaKaRa) according to a standard protocol: 95 °C for 10 min, followed by 40 cycles of 95 °C /15 s, 60 °C/20 s, and 72 °C /30 s, and ending with 72 °C for 5 min. In this study, the tomato *actin2* gene was chosen as a quantitative control. The relative gene expression was calculated using the 2^−ΔΔCt^ formula. A detailed description of all PCR primers used in this study is shown in Appendix A.

### 4.8. Analysis of Antioxidant Enzyme Activities, Malondialdehyde (MDA), Relative Water Content, and Root Vigor

To detect antioxidant enzyme activities and MDA levels under drought stress conditions, leaves at comparable developmental stages were collected. SOD, POD, CAT, and APX activities were analyzed according to established methods [77]. MDA content was detected as previously described [78]. A total of 0.3 g fresh leaf tissue was homogenized in 50 mM phosphate buffer (pH 7.8). The extract was centrifuged at 8000× *g* for 15 min, and the supernatant was added with 2.5 times volume thiobarbituric acid (TBA). Afterwards, the mixture was boiled for 20 min and then placed in the ice-bath. The sample was centrifuged for 5 min at 10000× *g* and the absorbance spectra of supernatant at wavelength 532 nm and 600 nm was monitored. Relative water content (RWC) was measured as previously described [79]. Fresh weight (FW) of tomato leaf was excised from the plants and immediately recorded. The leaf was soaked in distilled water at room temperature with constant light for 4 h, and the turgid weight (TW) was recorded. Dry weight (DW) was recorded after drying at 80 °C for 24 h. RWC was determined using the following formula: RWC (%) = [(FW − DW)/(TW −DW)] × 100. Root vigor was determined according to the triphenyltetrazolium chloride (TTC) method [80]. A total of 0.5 g fresh root tissue was soaked with 5 mL 0.4% *v*/*v* TTC and 5 mL 0.06 mol∙L^−1^ phosphate buffer (pH 7.0) and incubated in darkness at 37 °C for 3 h. The reaction was terminated by adding 2 mL of 1 mol L^−1^ sulfuric acid. The mixture was subsequently centrifuged and absorbance of supernatant was measured at wavelength 485 nm. Root vigor was calculated as the quantity of TTC reduced per gram of root sample per hour (μg g^−1^ h^−1^).

### 4.9. Statistical Analysis

All experiments were repeated three times under the same conditions, and significant differences were analyzed by Tukey’s studentized range (HSD) test (*p* < 0.05).

## Figures and Tables

**Figure 1 ijms-20-06271-f001:**
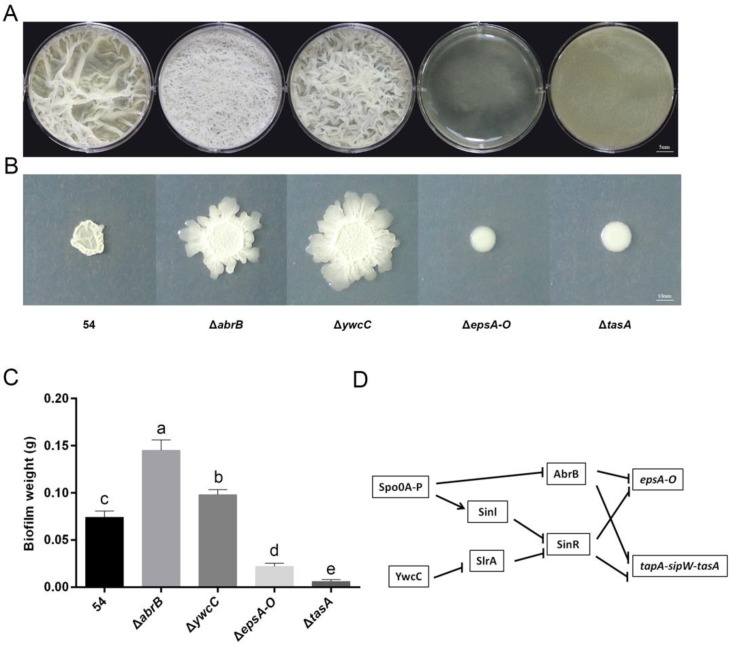
Pellicle development and colony formation of *Bacillus amyloliquefaciens* 54 wild strain, its hyper-robust and defective biofilm mutants. Cells were inoculated in LBGM liquid medium (**A**) for floating pellicle development and LBGM solid media (**B**) for colony formation. (**C**) Biofilm dry weight of the same strains as in (**A**) were evaluated. Data represent the mean ± SD (*n* = 3). Different letters on top of the bars indicated significant differences among different treatments. Tukey’s studentized range (HSD) test was applied in the statistical analysis. (**D**) The genetic circuitry that regulates biofilm formation in *Bacillus* spp. Each photograph was taken 36 or 48 h of post-incubation at 30 °C.

**Figure 2 ijms-20-06271-f002:**
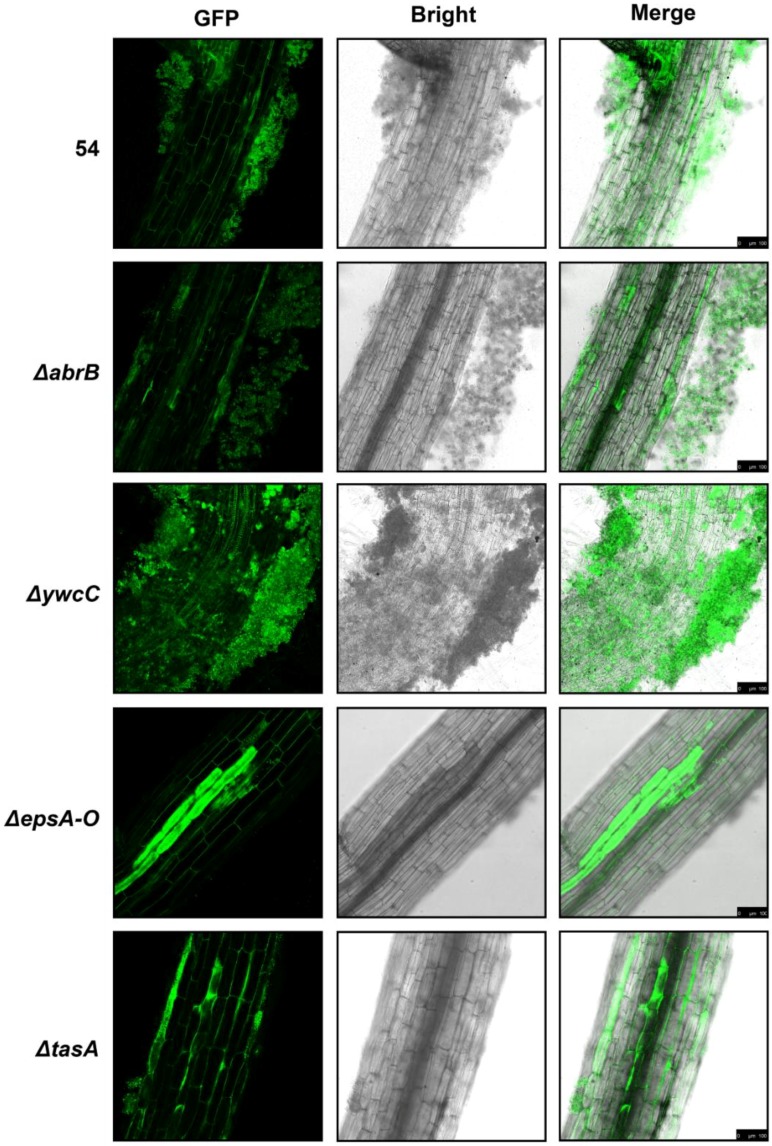
Biofilm formation on the surfaces of tomato root by *B. amyloliquefaciens* 54 wild strain, its hyper-robust and defective biofilm mutants. Biofilm formation of the wild type *B. amyloliquefaciens* 54 (in the top panels), two hyper-biofilm mutants (Δ*abrB* and Δ*ywcC*) (in the second and third panels) and two defective biofilm mutants (Δ*epsA-O* and Δ*tasA*) (in the fourth and bottom panels) are shown. Green lines shown in the figure are *B. amyloliquefaciens* 54 wild strain and its derivatives which labled by GFP protein. The scale bar is 100 μm.

**Figure 3 ijms-20-06271-f003:**
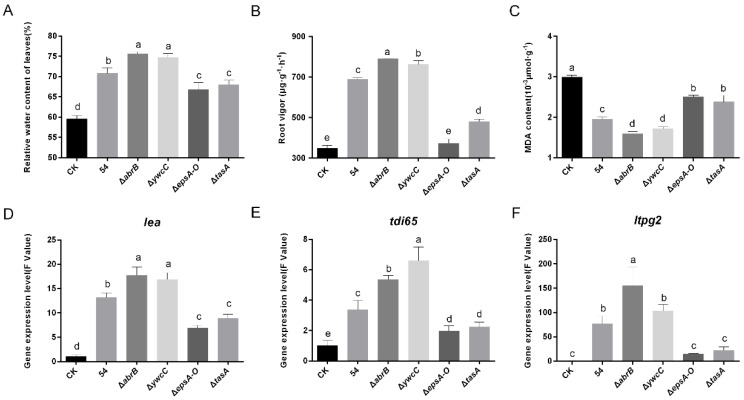
Biofilm formation is positively involved in *B. amyloliquefaciens* 54-mediated physiological indicators of tomato drought tolerance and stress-responsive genes expression. Changes of relative water content (RWC) content in leaves (**A**), root vigor (**B**), malondialdehyde (MDA) content (**C**). The relative transcript levels of stress-responsive genes *lea* (**D**), *tdi65* (**E**), *ltpg2* (**F**). *SlActin* was used as an internal control to normalize individual gene expression values. The data were measured after microbe inoculation and subsequently withholding water for 11 days. Data represent the mean ± SD (*n* = 3). Different letters on top of the bars indicated significant differences among different treatments. Tukey’s studentized range (HSD) test was applied in the statistical analysis.

**Figure 4 ijms-20-06271-f004:**
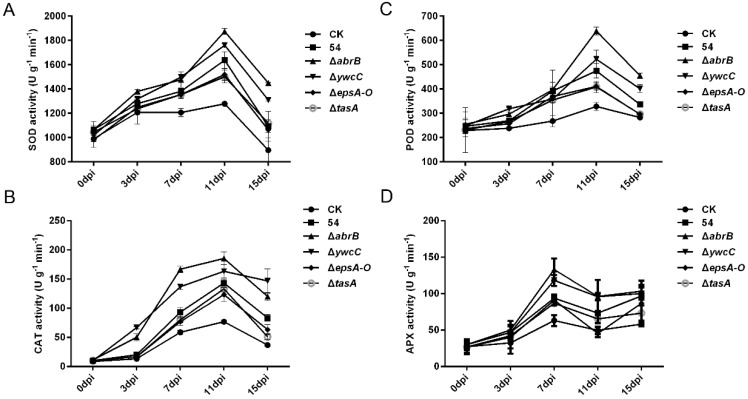
Biofilm formation is positively involved in *B. amyloliquefaciens* 54-mediated antioxidant enzymes activities. Changes of SOD activity (**A**), POD activity (**B**), CAT activity (**C**), and APX activity (**D**) were measured after microbe inoculation under drought stress conditions. Standard errors of four independent samples are presented by the error bars. All experiments were repeated three times with similar results.

**Figure 5 ijms-20-06271-f005:**
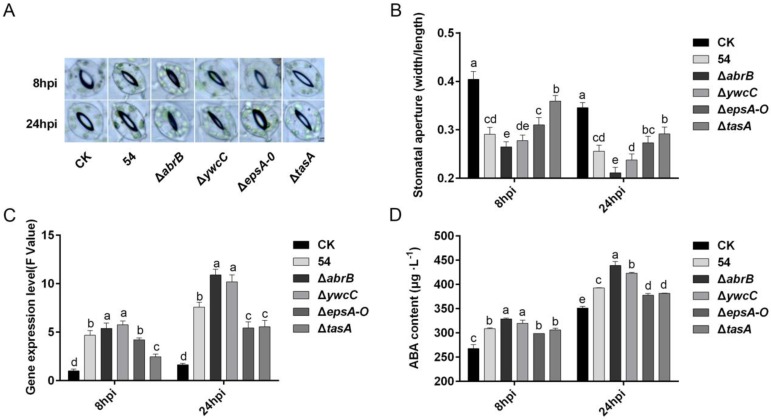
Biofilm formation contributes to *B. amyloliquefaciens* 54 mediating the closure of stomata through abscisic acid (ABA) pathways. Micrographs of stomata (**A**), stomatal aperture sizes (**B**), relative transcript levels of ABA biosynthetic gene *nced1* (**C**), total ABA content (**D**) in leaves of tomato plants were evaluated 8 h and 12 h post-incubation with LB medium (CK), wild strains *B. amyloliquefaciens* 54, and its derived mutants. *SlActin* was used as an internal control to normalize *nced1* expression value. Data represent the mean ± SD (*n* = 3). Different letters on top of the bars indicate significant differences among different treatments. Tukey’s studentized range (HSD) test was applied in the statistical analysis.

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
