# Peer review of "Biofilms Positively Contribute to Bacillus amyloliquefaciens 54-induced Drought Tolerance in Tomato Plants"

_ijms, 2019, doi:10.3390/ijms20246271_

Round 1

Reviewer 1 Report

Review on the publication by Wang et al under the title Biofilms Positively Contribute to Bacillus amyloliquefaciens 54-induced Drought Tolerance in Tomato Plants.

I have only minor comment to the manuscript:

- The author represent interesting peace of work. In my opinion authors have to add conclusion part into the manuscript.

Line 244 Timmusk et al (2015), please correct it and put the number instead of year of the publication. The same story with the lines 245, 317

Please revise the References as sometime you have the name of the journals without dots like for example in the citation number 7 and sometime with, for example, in the citations number 3, 9 and so on.

Authors Contribution – according to journal requirement you have to put only the first letter of the surname

Author Response

Dear Reviewer:

Thank you for your kind comments concerning our manuscript entitled “Biofilms Positively Contribute to Bacillus amyloliquefaciens 54-induced Drought Tolerance in Tomato Plants”. Your comments are all valuable and very helpful for revising and improving our paper, as well as the important guiding significance to our research. We have studied comments carefully and have made correction, mark it in red words, which we hope meet with approval.

Line 244. According to the IJMS guides, I have corrected it (Line 248). Also the same modification with lines 249, 321. You can easily find them in red words.

Indeed, I found some mistakes in the References part. I have unified the format of the reference. Except for the 3 mistakes you put forward, I also found 2 mistakes in 52 and 80. And I have modified them. According to the Reference Formatting Guides of IJMS, cited journals should be abbreviated according to ISO 4 rules. So, the abbreviated words should be added dots. But some words, such as “plant” and “cell” should not be abbreviated and without dots. 

Authors Contribution – according to journal requirement I have put only the first letter of the surname.

Best regards,

Jianhua Guo

Reviewer 2 Report

Authors have depicted the role of Bacillus amyloliquefaciens in biofilms formation during drought stress in tomato in this present study. The manuscript is relatively well-written and well-performed. However, some flaws that can be corrected to improve the understanding the manuscript.

Manuscript introduction section should be include few more line emphasizing on Bacillus spp. and regarding their role. In section 4.1, line 323; Please mentioned the source of bacterial strains used for study (If possible, include their accession no. also).

Author Response

Dear Reviewer:

Thank you for your kind comments concerning our manuscript entitled “Biofilms Positively Contribute to Bacillus amyloliquefaciens 54-induced Drought Tolerance in Tomato Plants”. Your comments are all valuable and very helpful for revising and improving our paper, as well as the important guiding significance to our research. We have studied comments carefully and have made correction, mark it in red words, which we hope meet with approval.

According to your suggestion, I have added some more content to introduce Bacillus spp. and their role in the Introduction part.

All the PGPR that we used in this study came from our biocontrol bacteria library, which consisted of beneficial bacteria isolated by our lab from the field. Among them, 54 was selected which has the best ability to induce drought tolerance in tomato plants. Except that, it has been reported to control bacterial fruit blotch (BFB) caused by Acidovorax avenae subsp. citrulli (Jiang et al., 2015).

Best regards,

Jianhua Guo